# HPV Vaccination among Polish Adolescents—Results from POLKA 18 Study

**DOI:** 10.3390/healthcare10122385

**Published:** 2022-11-28

**Authors:** Michalina Drejza, Katarzyna Rylewicz, Maria Lewandowska, Katarzyna Gross-Tyrkin, Grzegorz Łopiński, Joanna Barwińska, Ewa Majcherek, Klaudia Szymuś, Patrycja Klein, Katarzyna Plagens-Rotman, Magdalena Pisarska-Krawczyk, Witold Kędzia, Grażyna Jarząbek-Bielecka

**Affiliations:** 1Department of Obstetrics and Gynaecology, Cambridge University Hospitals NHS Foundation Trust, Cambridge CB2 0QQ, UK; 2Faculty of Medicine, Medical University of Warsaw, 02-091 Warsaw, Poland; 3Faculty of Public Health and Policy, London School of Hygiene & Tropical Medicine, London WC1E 7HT, UK; 4Department of Obstetrics and Gynaecology, District Hospital of Kartuzy, 83-300 Kartuzy, Poland; 5Faculty of Medicine, Medical University of Silesia, 40-055 Katowice, Poland; 6Faculty of Medicine, Poznan University of Medical Sciences, 61-701 Poznan, Poland; 7Faculty of Medicine, Medical University of Gdansk, 80-210 Gdansk, Poland; 8Faculty of Medicine, Medical University of Lublin, 20-059 Lublin, Poland; 9Center for Pediatric, Adolescent Gynecology and Sexology Division of Gynecology, Department of Perinatology and Gynecology, Poznan University of Medical Sciences, 61-701 Poznan, Poland; 10Department of Nursery, The President Stanislaw Wojciechowski Calisia University, 62-800 Kalisz, Poland; 11Department of Perinatology and Gynaecology, Poznań University of Medical Sciences, 61-701 Poznań, Poland

**Keywords:** HPV, cervical cancer, adolescent health

## Abstract

Human Papillomavirus (HPV) is the main cause of cervical cancer and genital warts and constitutes one of the most common sexually transmitted infections. Cervical cancer is the only reproductive cancer that has a primary prevention programme through the introduction of HPV vaccinations. Even though the majority of European countries have nationally funded HPV vaccination programmes, in Poland these are exclusively local and scarcely funded. Moreover, the majority of local programmes are directed to females only. Meanwhile, Poland has one of the highest cervical cancer incidence rates among high income countries. The aim of this study was to measure HPV vaccination levels among final-year students in Poland and to establish the association between vaccination status and gender, region and level of sexual education received. This study is a part of the POLKA 18 Study, which used original self-reported paper-based questionnaires distributed in schools in six Polish regions. The study was conducted between April and December 2019. The obtained data were analysed in STATA 17. In total, 2701 fully completed questionnaires were collected. Over half of the respondents (58.2%) did not know their HPV vaccination status. Only 16.0% of the respondents replied that they have been vaccinated against HPV (18.2% of females and 14.5% of males). There was no direct association between vaccination status and access to ‘family life education’ classes. The vaccination level significantly differed among the different regions of Poland (*p* < 0.0001), with the Śląskie and Wielkopolskie regions achieving the highest rates. At least a quarter of adolescents after their sexual debut have not been vaccinated against HPV. Regions with immunization programmes introduced to their provincial capitals had higher vaccination rates. Our findings indicate the need for the introduction of state-funded vaccination programmes at the national level for the vaccination rate to increase, which will have the potential to decrease cervical cancer incidence in the country.

## 1. Introduction

Human papillomavirus (HPV) belongs to the family of *Papillomaviridae* and is the most common sexually transmitted infection in the world, which infects the majority of sexually active people in their lifetime [1,2,3]. HPV is susceptible to both cutaneous and mucosal epithelia; thus, it can cause lesions in various places of the body, including the skin, anogenital tract and oral cavity. However, most HPV infections clear by themselves within two years [3,4]. Importantly, the vast majority of cervical cancers are attributed to long-lasting HPV infections, and 70% of them are caused by two types of HPV—16 and 18 [5]. HPV poses a risk not only to women; it is also highly prevalent in men and can lead to warts as well as penile, anal or oropharyngeal cancers [6,7]. Sexually active men also play an important role in transmitting the virus to their female partners, contributing to higher HPV transmission and maintenance of infection [8].

Cervical cancer is the fourth-most-commonly diagnosed cancer in women and the fourth leading cause of cancer mortality in women [9]. Each year there are around 33,000 new cases diagnosed in the European Union and around 15,000 cervical-cancer-related deaths [10]. Nevertheless, its incidence and mortality have drastically fallen over the last 30–40 years due to successful screening, significant increase in the quality of treatment and the emergence of an effective HPV vaccine. Despite its continued high prevalence, HPV infection is nowadays almost completely preventable owing to HPV vaccinations (primary prevention method) and screening by performing a cervical smear (secondary prevention) [11].

In contrast, Poland is yet to introduce nationwide primary prevention of cervical cancer, in spite of the alarming evidence of the country’s high incidence and high mortality of cervical cancer; the latter measure in Poland was estimated to 7.9 per 100,000 women, compared to the European 2.8 average. Poland has established a national secondary prevention of cervical cancer programme, which is directed to women aged 25–59, who can undergo a screening cytology test free-of-charge every 3 years [12]. The participation in routine HPV screenings, primarily based on cytology, is also alarmingly low—18.2% compared to over 60% in Finland, Netherlands and Denmark. Between 2000 and 2015, 28,274 women in Poland died of cervical cancer [13,14]. 

There are three different vaccines against HPV available in the European Union, which differ in the number of HPV types they protect against—a bivalent, a quadrivalent and a nine-valent vaccine. Multiple positive effects have been observed after the introduction of HPV vaccinations, including the lower incidence of the cancerogenous HPV 16 and 18 infections, as well as those of HPV 6 and 11, which cause genital warts. Moreover, HPV vaccination programmes have been shown to lead to a decrease in cytological low-grade cervical abnormalities and histologically proven cervical abnormalities [15,16].

Currently, HPV vaccinations in Poland are included in the list of recommended vaccinations, especially for adolescents and people before sexual-life initiation. Even though in 2021, the Agency for Health Technology Assessment and Tariff System issued a recommendation for the reimbursement of four-valent vaccinations for children over 9 years of age, no national HPV vaccination programme has been established to date [17]. The only nationally partially funded vaccine is Cervarix, a bivalent vaccine against HPV types 16 and 18. Since 1 November 2021, patients are able to buy the vaccine for a 50% reduced fee in all registered indications and recommendations. The cost of the whole treatment to individual patients is still high, around 415 PLN (c. 88 EUR), which is over one-tenth of the minimum wage [18]. Gardasil 4, a four-valent vaccine, and Gardasil 9, a nine-valent vaccine, are both available commercially for the full price of around 900 PLN (c. 191 EUR, which is three-tenths of the minimum wage) and 1650 PLN (c. 350 EUR, over one-half of the minimum wage) for the whole treatment, respectively [19]. There are several regional programmes reimbursing the vaccines for adolescents over the age of 9 as part of the local vaccination schemes. The majority of vaccination programmes are directed to females only, but a few also accept male adolescents [20]. 

Poland has no comprehensive sexual education programmes, and the ‘family life education’ classes have no mention of HPV or the vaccines against it in its core curriculum [21]. There have already been studies showing that Polish parents’ knowledge on HPV is low [22]. Little is known about the knowledge and attitudes of those who did obtain the vaccine. 

Given the lack of a national vaccination programme, in this study we aimed to estimate the vaccination coverage among final-year high school students across different regions of Poland and across genders. We also aimed to assess whether there were differences in vaccination levels dependent on the level of sexual education received by the adolescent. With the understanding of whether sexual education is a contributing factor to HPV vaccination coverage, we can best design interventions that would increase the vaccination uptake before a national programme is developed and after one is implemented.

## 2. Materials and Methods

This paper is a part of wider research, the POLKA 18 cross-sectional study investigating Polish adolescents’ knowledge, attitudes and practices within the domain of sexual and reproductive health. The study was based on original, self-reported, paper-based questionnaires.

Before the POLKA 18 questionnaire development process, the team conducted local consultations with young people in secondary schools, as well as with experts and academics, to create a model of health priorities for Polish adolescents. Their ideas and perceived priorities were used in creating the survey. 

The study was divided into two phases—the pilot phase and phase II. Both were based on original, self-reported, paper-based questionnaires. They were distributed in high schools and vocational schools by medical students associated with the International Association of Medical Students—IFMSA-Poland. The schools had to give consent to participate in the study in order for the questionnaires to be distributed. The majority of the schools were chosen based on previous cooperation with IFMSA-Poland and some were newly recruited after offering educational activities provided by the student organisation. We approached final-year students in high schools and also penultimate-year students in vocational schools only to ensure they were over the age of 18 and would not need parental consent for participation. Adolescents who participated in the study were primarily aged 18–19; some were over 19 but were still in high school. The questionnaires were filled in during classes in which IFMSA-Poland representatives were conducting educational activities. Each student present during class had the opportunity to fill in the questionnaire. On the front page of the questionnaire, the participants were informed about the subject matter of the study, assured of anonymity, asked not to write down any personal data and informed about their right of refusal to participate in the study. The contact details of the research team were provided as well. 

The questionnaires were distributed in six Polish regions (voivodeship): Mazowieckie, Śląskie, Wielkopolskie, Lubelskie, Pomorskie and Zachodniopomorskie. The pilot phase did not include the Zachodniopomorskie region; thus, the data from this voivodeship have been combined with the data from the Pomorskie region to maintain the weighing and disaggregation of the data.

The data used in this study regarding age and gender, region of living, HPV vaccination status and sexual debut, as well as sexual education availability and attendance, were retrieved from the POLKA 18 survey results for the purpose of this analysis. The region of living was precoded on the questionnaire by the research assistants. The rest of the questions can be found in Appendix A. The obtained data were cleaned by scanning the records for missing and incorrect entries for each variable. Chi-square tests were used to assess the evidence for association between the exposure variables (demographic, sexual education and sexual debut variables) and the outcome of HPV vaccination status. *p*-values for differences in those proportions across categories were reported. The analysis was carried out using STATA 17. 

Even though in the question about gender we had the option of ‘other’, given the very low numbers of participants who reported so, we used binary gender categories for the statistical tests. Additionally, the place of living referred to the place of residence and not the place of study.

The study received ethical approval from the Ethics Committee of Poznan University of Medical Sciences (13 December 2018).

## 3. Results

### 3.1. General Characteristics of the Study Group

In total, 2701 fully completed questionnaires were collected and included in the final analysis of the study. The general characteristics of the sample can be found below in Table 1. In Table 2, the sample was characterised by vaccination status.

Overall, the number of questionnaires collected in each region of Poland were as follows: 737 in Mazowieckie, 653 in Wielkopolskie, 518 in Śląskie, 259 in Lubelskie and 533 in Pomorskie and Zachodniopomorskie voivodeships. 

The respondents in the study had the option of marking three responses in terms of gender identity. In the study, women constituted 58.2% (*n* = 1572), men—40.6% (*n* = 1096) and other—0.8% (*n* = 20). Moreover, 0.4% (*n* = 13) did not mark their answers for this question. 

The study population was represented almost equally by adolescents living in cities with more than 100,000 inhabitants (32.7%), cities with fewer than 100,000 inhabitants (29.9%) and in rural areas (36.7%).

### 3.2. HPV Vaccination Level in Poland by Gender

All respondents were asked whether they were vaccinated against HPV. Only 16.0% (*n* = 432) of the respondents replied that they had been vaccinated, 22.3% (*n* = 602) said that they had not been vaccinated, and 58.2% (*n* = 1571) did not know whether or not they had been vaccinated. Among females, 18.2% reported having been vaccinated, and among males, 14.5%; however, the difference across genders was not significant (*p* = 0.106).

### 3.3. HPV Vaccination Level in Different Regions of Poland

The number of vaccinated adolescents differed between voivodeships, with the highest rate of 23.0% vaccinated adolescents in Śląskie, a slightly lower 20.4% (22.4% females vs. 17.2% males) in the Wielkopolskie, 16.5% in Lubelskie, 12.0% in the Mazowieckie, and the lowest rate of 11.9% in Pomorskie and Zachodniopomorskie voivodeships. There was strong evidence that the vaccination level varied significantly across the different regions of Poland (*p* < 0.001). The level of HPV vaccination in Poland among adolescents in the POLKA 18 study across regions and stratified by gender can be found in Figure 1. 

There was a significant difference (*p* = 0.005) observed in the vaccination rate among adolescents living in different types of urban areas. Among adolescents living in cities with over 100,000 inhabitants, 19.8% were vaccinated, in cities with less than 100,000 inhabitants—15.7% and in rural areas—14.5%. 

### 3.4. HPV Vaccination Level Depending on Sexual Education at Schools

Students were asked about the sexual education classes being organized and attended at their schools. The study looked into associations between the vaccination level and attendance and availability of those classes at schools. The students who had the ‘family life education’ classes available at school tended to have higher vaccination levels (17.4% vs. 15.0%; *p* = 0.018). There was no statistical difference (*p* = 0.761) between students who attended sexual education classes at school (16.0%) and those who did not (16.6%). 

### 3.5. HPV Vaccination Level by Sexual Debut

Among students who reported having had intercourse, only under one-fifth (19.8%) reported having been vaccinated. Over one-quarter (26.8%) were not vaccinated, and over half did not know their vaccination status. The proportion of those vaccinated was lower in the adolescents who had not had their sexual debut (13.2%). There was strong evidence that the differences were statistically significant (*p* < 0.001).

## 4. Discussion

### 4.1. Main Findings

We found that fewer than one-fifth of the adolescent population in Poland had been vaccinated against HPV. The European Centre for Disease Prevention and Control considers the vaccination coverage rate below or equal to 30% to be very low, and Poland has been consistently placed in this category together with three other European countries—Bulgaria, France and Greece [15]. This is likely a result of the lack of a structured, free-of-charge, national vaccination programme, an intervention evidenced as leading to higher vaccination rates in other European countries, together with sending invitations and reminders, as well as relying on schools as the place of vaccine administration [15]. The majority of the participants in our study were not aware whether they had received the vaccination or not at all, which suggests an extremely low level of knowledge on HPV and cervical cancer prevention in general. However, it might also mean that some of the students who did not know their vaccination status might have actually received the vaccine—therefore, the true vaccination rate could be slightly higher. This calls for the HPV vaccination programmes in Poland to record vaccinations systematically and on a national level. 

The lack of statistical difference between females and males was a surprising finding as the majority of local vaccination programmes in Poland include only females [20]. However, the possibility to vaccinate one’s child out-of-pocket in private healthcare settings is a potential reason behind the lack of statistical difference. The potential lack of knowledge regarding the participants’ own HPV vaccination status was likely to have introduced some bias, as a proportion of these respondents could have been vaccinated and not been aware of it. Equally, some adolescents might have wrongly assumed that it is an obligatory vaccine within the national immunisation programme and therefore answered as if they had been vaccinated as well. However, our finding that at least a quarter of adolescents who had had a sexual debut (and therefore have likely been exposed to HPV) had not been vaccinated against the virus is a striking piece of evidence showing the missed potential of the vaccine programmes.

There were conflicting results when it came to sexuality education classes and vaccination coverage rates. There were higher vaccination rates at schools where ‘family life education’ classes were taught. However, there was no evidence that individual attendance of those had any association with vaccine coverage. These contrasting findings likely indicate district-level differences in vaccine programmes, their uptake differing more on the local rather than the individual level. However, the strikingly low knowledge about the participants’ own HPV vaccination status, irrespective of their sexual education, suggests that the noncomprehensive sexuality education does not cover the substantial topics of HPV and cancer prevention sufficiently, and that there is not enough coverage of those topics in popular media to increase awareness among young people.

### 4.2. Strengths and Limitations

The strengths of this study were its relatively large sample size and its inclusion of participants from various regions and from both urban and rural areas. It was a part of a comprehensive questionnaire regarding the sexual and reproductive health of young people. To our knowledge, it is the first one to be carried out in Poland. 

The most important limitation of the study was the sampling, which was not nationally representative, but based on cooperation between the schools and the study team. Consequently, the recruitment was opportunistic, and we were not able to calculate the response rate. The subject of sexual health and sexuality is still highly stigmatised in Poland and carrying out research in adolescent populations across schools throughout the country would have likely been faced with opposition. Another limitation of the study is the fact that the questionnaires were based on self-reporting, which could have introduced bias because some questions might have been not understood by all students in the same way (for instance, depending on their definition of sexual intercourse). Because the study was carried out at schools, some students might have also not answered truthfully in order to impress their peers—however, we cannot estimate the scale of that phenomenon. The team carried out the surveys in a professional and confidential manner to minimize that risk. 

### 4.3. Implications for Policy, Practice and Further Research

Regional and local vaccination programmes have been shown to improve the general vaccination coverage rate. Regions with immunisation programmes being introduced to their provincial capitals—Poznan (Wielkopolskie) and Katowice (Slaskie) [23] in earlier years—had significantly higher vaccination rates in our study compared to other regions. The differences are likely caused by the existence of local vaccination programmes in some regions, and their absence in others. However, even in these regions the vaccination rate has not surpassed 30%, and thus was still considered to be very low. It is a strong suggestion that the local programmes do not bring sufficient results. This points towards the need for the introduction of fully subsidised vaccines in the national immunisation programme in order to make them accessible and easier to coordinate with current compulsory vaccinations, and hopefully more acceptable to the general public. 

The current vaccination programmes are ongoing in both big cities and smaller towns. However, they are focused in the most-urbanised areas, which could be a potential reason behind the significant difference between cities with over 100,000 inhabitants, cities with below 100,000 inhabitants and villages. However, a potential limitation of the study is also the fact that the adolescents were asked about the place of living and not the place of study. Moreover, it is important to remember that not all localities have ongoing local vaccination programmes, which can create disparities in vaccination access across counties, regardless of the number of inhabitants. That is one example of how the lack of a national programme furthers health inequity, as adolescents from more liberal municipalities are more likely to receive the vaccine than those from the more conservative one, consequently achieving better health outcomes. 

It is very challenging to obtain nationwide-level data in Poland on HPV vaccination rates, given that the programmes are fragmented and local. Introducing a fully funded national vaccination programme and register would allow not only appropriate data collection for monitoring and evaluation, but also track its impact on the eradication of HPV and HPV-related cancers. 

Moreover, the attitudes of society towards vaccines in general, but especially in the context of the prevention of sexually transmitted infections, are still likely making some parents reluctant with regards to allowing their children to be vaccinated against HPV. Therefore, it might be advisable to create educational interventions not only towards adolescents themselves, but also the parents as enablers for adolescent healthcare. According to recent studies [24], the attitudes of the parents with regards to their child being vaccinated against HPV are mostly positive; however, the overall knowledge about HPV in this group is low. Most of the significant factors that influenced their willingness were modifiable, such as raising awareness about HPV and having positive attitudes toward vaccines in general. More research should be conducted on how awareness-raising campaigns, including through social media, can be most effective in improving attitudes and vaccine uptake.

## 5. Conclusions

The coverage of the HPV vaccine in Poland is among the lowest in Europe. Regions with immunisation programmes introduced to their provincial capitals had higher vaccination rates in our study. However, even there the vaccination coverage rate is considered to be very low. This indicates the need for a nationally funded HPV vaccination programme in order for the vaccine coverage to increase to satisfactory levels with the capacity of cancer and other HPV-related disease prevention. At least a quarter of adolescents after their sexual debut have not been vaccinated against HPV, which shows a great missed potential of the vaccine programmes. Sexuality education, where it is delivered at schools, was not found to improve HPV vaccine coverage nor awareness—therefore indicating a need for a comprehensive curriculum, which would provide adolescents with the necessary knowledge enabling them to make informed decisions about their sexual and reproductive health, including the importance of HPV prevention.

## Figures and Tables

**Figure 1 healthcare-10-02385-f001:**
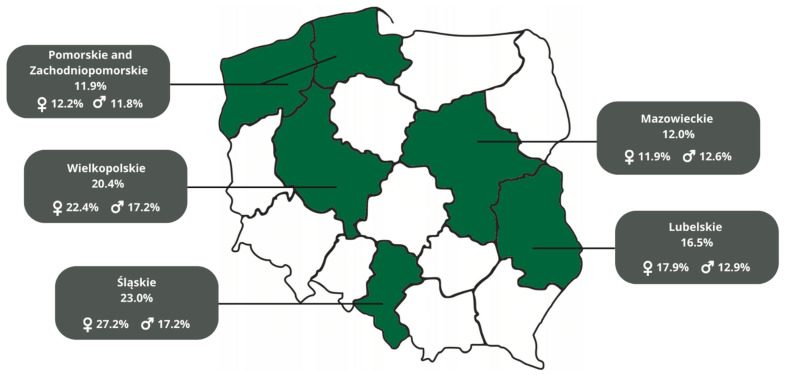
HPV vaccination level in Poland among adolescents in the POLKA 18 study.

**Table 1 healthcare-10-02385-t001:** General characteristics of the sample (*n* = 2701) (%).

HPV Vaccination Status *	
Vaccinated	432 (16.0%)
Unvaccinated	602 (22.3%)
Do not know	1571 (58.2%)
**Age** ** ^1^ **	
18	2021 (74.8%)
19	515 (19.1%)
Over 19	159 (5.9%)
**Gender** ** ^2^ **	
Woman	1572 (58.2%)
Man	1096 (40.6%)
Other	20 (0.8%)
**Residence ^3^**	
City over 100,000	883 (32.7%)
City under 100,000	808 (29.9%)
Rural area	991 (36.7%)
**Region of Poland ^4^**	
Śląskie	518 (19.2%)
Mazowieckie	737 (27.3%)
Wielkopolskie	653 (24.2%)
Lubelskie	259 (9.6%)
Zachodniopomorskie i Pomorskie	533 (19.7%)
**Sexual orientation ^5^**	
Heterosexual	2196 (86.9%)
Homosexual	63 (2.5%)
Bi/pansexual	201 (8.0%)
Asexual	30 (1.2%)
Other	37 (1.5%)
**Sex education ^6^**	
School offers classes	1512 (56.0%)
School doesn’t offer classes	1121 (41.5%)
**Sex education attendance ^7^**	
Attended	1406 (52.1%)
Did not attend	1236 (45.8%)
**Sexual debut ^8^**	
Yes	1382 (51.2%)
No	1269 (47.0%)

Note: where the percentages in row do not sum up to a 100, it is due to observations missing as follows: * 96 observations missing, ^1^ 6 observations missing, ^2^ 13 observations missing, ^3^ 19 observations missing, ^4^ 1 observation missing, ^5^ 174 observations missing, ^6^ 68 observations missing, ^7^ 59 observations missing, ^8^ 50 observations missing.

**Table 2 healthcare-10-02385-t002:** Characteristics of the sample by vaccination status (*n* = 2605) (%) * (*p*-values from chi-square tests for difference in proportion of HPV vaccination across ages, genders, residences, regions, sex education, sexual debut).

	Vaccinated	Unvaccinated	Do Not Know	*p*-Value **
**Age**				0.721
18	325 (16.6%)	428 (21.9%)	1206 (61.6%)	
19	77 (15.6%)	131 (26.6%)	285 (57.8%)	
Over 19	20 (19.6%)	41 (27.7%)	78 (52.7%)	
**Gender**				0.106
Female	278 (18.2%)	358 (23.4%)	896 (58.5%)	
Male	151 (14.5%)	236 (22.6%)	656 (62.9%)	
Other	2 (11.1%)	4 (22.2%)	12 (66.7%)	
**Residence**				0.005
City over 100,000	169 (19.8%)	203 (23.8%)	481 (56.4%)	
City under 100,000	122 (15.7%)	191 (24.6%)	464 (59.7%)	
Rural areas	139 (14.5%)	205 (21.4%)	614 (64.1%)	
**Region of Poland**				<0.001
Śląskie	117 (23.0%)	117 (23.0%)	275 (54.0%)	
Mazowieckie	85 (12.0%)	171 (24.2%)	451 (63.8%)	
Wielkopolskie	128 (20.4%)	131 (20.9%)	368 (58.7%)	
Lubelskie	41 (16.5%)	60 (24.1%)	148 (59.4%)	
Pomorskie i Zachodniopomorskie	61 (11.9%)	123 (24.0%)	329 (64.1%)	
**Sex education**				0.018
School offers classes	256 (17.4%)	318 (21.6%)	897 (61.0%)	
School does not offer classes	162 (15.0%)	282 (26.2%)	634 (58.8%)	
**Sex education attendance**				0.761
Attended	219 (16.0%)	327 (32.9%)	821 (60.1%)	
Did not attend	198 (16.6%)	271 (22.8%)	722 (60.6%)	
**Sexual debut**				<0.001
Have had	268 (19.8%)	362 (26.8%)	721 (52.4%)	
Have not had	163 (13.2%)	234 (18.9%)	839 (67.9%)	

* only entries with nonmissing values for HPV vaccination status included. ******
*p*-value from chi-square test for difference in proportions.

## Data Availability

Not applicable.

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
