# Peer review of "HPV Vaccination among Polish Adolescents—Results from POLKA 18 Study"

_healthcare, 2022, doi:10.3390/healthcare10122385_

Round 1

Reviewer 1 Report

Dear authors,

 The submission addressed HPV vaccination knowledge in the students from high school and vocational college by questionnaire survey. The survey included age,gender,living place, sexual education, attendance and vaccination status.The data presented in the investigation disseminates demerits of HPV vaccination program in Poland.

 However, the manuscript is not satisfied for the acceptance before minor revision and the comments are listed as the followings:

1. In the M& M section, please provides an ethic approval code or number from university or institute.

2. In the Discussion section, parent knowledge of HPV prevention and social media might have a great impact on male/female option of vaccination. Authors have to input above influences in the discussion.

3.Grammatic mistakes should be improved across the whole manuscript by a native language speaker.

Author Response

Dear Reviewer, 

Thank you for your in-depth reading of our work and for all your comments. 

Please see below for our responses to all the points raised:

  1. In the M& M section, please provides an ethic approval code or number from university or institute.

There is no specific code for the ethical approval, we are happy to provide a scan of the document with the date if required. 

  1. In the Discussion section, parent knowledge of HPV prevention and social media might have a great impact on male/female option of vaccination. Authors have to input above influences in the discussion.

In the discussion section, we wrote: “Moreover, the attitudes of society towards vaccines in general, but especially in context of prevention of sexually transmitted infections, is still making some parents likely to be reluctant with regards to allowing their children to be vaccinated against HPV. Therefore, it might be advisable to create educational interventions not only towards adolescents themselves, but also the parents as enablers for adolescents’ healthcare. More research should be done how awareness-raising campaigns, including via social media, can be most effective in improving attitudes and vaccine uptake.” 

3.Grammatic mistakes should be improved across the whole manuscript by a native language speaker.

Thank you, we have done so.

Reviewer 2 Report

The paper entitled "HPV vaccinations among Polish adolescents - results from POLKA 18 study" by Michalina Drejza et al. reports the evaluation of the HPV vaccination in Poland, based on the analysis of a series of 2701 fully completed questionnaires that could be collected. The study is well performed and encompasses a large number of cases. The result is unequivocal since it shows that only 18.2% of females and 14.5% of males were vaccinated against HPV. Interestingly, significant differences in HPV vaccination rates among the regions of Poland were observed. The reasons for the global low rate of vaccination are discussed and the paper is useful as a basis for the development of vaccination strategy in the country. My only remark is that there is a high rate (58,2%) of people that "did not know" if they had received or not the HPV vaccine. This might modify the real rate of HPV vaccination since a part of them could have received the vaccine. This pont could be discussed and the potential causes for the unability of young people to answer the question could also be discussed. In the perspective of development of the HPV vaccination in Poland, propositions should be made to record vaccination systematically. 

Author Response

Dear Reviewer, 

Thank you for your in-depth reading of our work and for all your comments. 

Please see below for our responses to all the points raised:

My only remark is that there is a high rate (58,2%) of people that "did not know" if they had received or not the HPV vaccine. This might modify the real rate of HPV vaccination since a part of them could have received the vaccine. This pont could be discussed and the potential causes for the unability of young people to answer the question could also be discussed. In the perspective of development of the HPV vaccination in Poland, propositions should be made to record vaccination systematically. 

We have added: “However, it might also mean that some of the students who did not know their vaccination status might have actually received the vaccine - therefore the true vaccination rate could be slightly higher. This calls for the HPV vaccination programmes in Poland to record vaccination systematically and on a national level. “

Reviewer 3 Report

No problems with content other than please check grammar and typos.

Author Response

Dear Reviewer, 

Thank you for your in-depth reading of our work and for all your comments. 

Please see below for our responses to all the points raised:

No problems with content other than please check grammar and typos.

Thank you, we have done so. 

Reviewer 4 Report

Thank you for the opportunity to review a manuscript entitled “HPV vaccinations among Polish adolescents - results from POLKA 18 study”. The manuscript presents a questionnaire survey analyzing the level of HPC vaccine coverage among Polish adolescents concerning their particular characteristics. The work is very well described in all details. The topic is correctly introduced, methods well characterized and the results are clearly presented. The study seems to be a bit regional—specific having beneficial potential mainly for Poland, however certain findings are generally valid and may be useful even in other countries.

I can see a few possible points to improve:

On the second line in the 5th paragraph of the introduction, "before sexual life initiation" would be better.

Please indicate the equivalent of the costs in PLN and also in EUR.

In the last paragraph of the introduction, “vaccination coverage” instead of “vaccination levels” would fit better.

In the methods, “The obtained data was cleaned…” This is too general. It could be specified.

What was the response rate? (How many people were asked to participate?) – It could be stated at the beginning of the results.

In the discussion, despite being very practical and well-structured, I miss some confrontations with existing literature. 

Best Regards.

Author Response

Dear Reviewer, 

Thank you for your in-depth reading of our work and for all your comments. 

Please see below for our responses to all the points raised:

On the second line in the 5th paragraph of the introduction, "before sexual life initiation" would be better.

Changed as suggested. 

Please indicate the equivalent of the costs in PLN and also in EUR.

Added as suggested. 

In the last paragraph of the introduction, “vaccination coverage” instead of “vaccination levels” would fit better.

Changed as suggested. 

In the methods, “The obtained data was cleaned…” This is too general. It could be specified.

We have changed this paragraph to: “The obtained data was cleaned by scanning the records for missing and incorrect entries for each variable. Chi-square tests were used to assessed the evidence for association between the exposure variables (demographic, sexual education and sexual debut variables) and the outcome of HPV vaccination status and p-values for difference in those proportions across categories were reported. The analysis was carried out using STATA 17.”

What was the response rate? (How many people were asked to participate?) – It could be stated at the beginning of the results.

As the recruitment was opportunistic (survey carried out in schools affiliated with the study, who agreed to participate, and in classes of those teachers who consented to carry out the survey), we were not able to record overall response rates. We have added the following in the discussion: “Consequently, the recruitment was opportunistic and we were not able to calculate the response rate.”

In the discussion, despite being very practical and well-structured, I miss some confrontations with existing literature. 

Added references to existing literature on parent's attitudes towards HPV vaccination of adolescents. and parents' as enablers of vaccination. Otherwise we believe that discussion covers most of emerging issues in the literature which is being thoroughly set out in our paper.

Reviewer 5 Report

The term Papillomaviridae has to be in italics.

Please explain which statistical tests were used. Just mentioning the program used is not enough. What do you mean that the data was "cleaned"? What is the interval of confidence?

What are the pitfalls of self-reported and paper questionnaires? In other words, do you have any estimation of the percentage of subjects that do not answer these questionnaires truthfully?

The article only contains tables. It would be better also to include charts that could make the data more accessible. Also, the way the table is organized is very confusing.

In table 2. For which comparison are the p-Values? This is very confusing.

What was the percentage of subjects that did not answer the test? This data could be important in trying to understand the sociological phenomena that could affect vaccination.

Why are the vaccination levels in The Slaskie and Wielkopolskie regions higher than in the other areas?

Author Response

Dear Reviewer, 

Thank you for your in-depth reading of our work and for all your comments. 

Please see below for our responses to all the points raised:

The term Papillomaviridae has to be in italics.

Changed as suggested. 

Please explain which statistical tests were used. Just mentioning the program used is not enough. What do you mean that the data was "cleaned"? What is the interval of confidence?

We added as follows: “The obtained data was cleaned by scanning the records for missing and incorrect entries for each variable.” We reported p-values for each difference in proportion instead of reporting 95% confidence intervals. 

What are the pitfalls of self-reported and paper questionnaires? In other words, do you have any estimation of the percentage of subjects that do not answer these questionnaires truthfully?

We added: “ Another limitation of the study is the fact that the questionnaires were based on self-report, which could have introduced bias because some questions might have been not understood by all students in the same way (for instance depending on their definition of sexual intercourse). Because the study was carried out at schools, some students might have also not answered truthfully in order to impress their peers - however we cannot estimate the scale of that phenomenon. The team carried out the surveys in a professional and confidential manner to minimize that risk.  “ 

The article only contains tables. It would be better also to include charts that could make the data more accessible. Also, the way the table is organized is very confusing.

We have reorganized the table in a way that is hopefully more clear. Please do let us know if you think a specific chart would be useful to the readers. 

In table 2. For which comparison are the p-Values? This is very confusing.

Added in the heading of the table - these p-values are derived from the chi-squared tests for difference in proportion of HPV vaccination across categories for each variable. 

What was the percentage of subjects that did not answer the test? This data could be important in trying to understand the sociological phenomena that could affect vaccination.

As the recruitment was opportunistic (survey carried out in schools affiliated with the study, who agreed to participate, and in classes of those teachers who consented to carry out the survey), we were not able to record overall response rates. We have added the following in the discussion: “Consequently, the recruitment was opportunistic and we were not able to calculate the response rate.”

Why are the vaccination levels in The Slaskie and Wielkopolskie regions higher than in the other areas?

We added: “The differences are likely to be caused by the existence of local vaccination programmes in some regions, and their absence in others.”